# Downregulation of Mannose-6-Phosphate Receptors in Fabry Disease Cardiomyopathy: A Potential Target for Enzyme Therapy Enhancement

**DOI:** 10.3390/jcm11185440

**Published:** 2022-09-16

**Authors:** Andrea Frustaci, Romina Verardo, Rossella Scialla, Giulia Bagnato, Margherita Verardo, Maria Alfarano, Matteo A. Russo

**Affiliations:** 1Department of Clinical, Internal, Anesthesiologist and Cardiovascular Sciences, La Sapienza University, 00161 Rome, Italy; 2Cellular and Molecular Cardiology Lab, IRCCS L. Spallanzani, 00149 Rome, Italy; 3Unit of Neuromuscular and Neurodegenerative Disorders, Genetics and Rare Diseases Research Division, Bambino Gesù Children’s Hospital, 00146 Rome, Italy; 4MEBIC Consortium, San Raffaele Open University, 00166 Rome, Italy; 5Cellular and Molecular Pathology, IRCCS San Raffaele, 00166 Rome, Italy

**Keywords:** Fabry disease, GLA, Mannose-6-phosphate receptors, globotriaosylceramide, cardiomyocyte, enzyme replacement therapy

## Abstract

Background: The efficacy of enzyme replacement therapy (ERT) in mobilizing globotryaosylceramide (GB-3) from Fabry cardiomyocytes is limited. The mechanism involved is still obscure. Methods: Assessment of M6Pr, M6Pr-mRNA, and Ubiquitin has been obtained by Western blot analysis and real-time PCR of frozen endomyocardial biopsy samples, from 17 pts with FD, various degree of left ventricular hypertrophy, and maximal wall thickening (MWT) from 11.5 and 20 mm. The diagnosis and severity of FDCM followed definitions of GLA mutation, α-galactosidase A enzyme activity, cardiac magnetic resonance, and left ventricular endomyocardial biopsy with the quantification of myocyte hypertrophy and the extent of Gb-3 accumulation. All patients have received alpha or beta agalsidase for ≥3 years without a reduction in LV mass nor an increase in T1 mapping at CMR. Controls were surgical biopsies from 15 patients undergoing mitral valve replacement. Results: Protein analysis showed mean M6Pr in FDCM to be 5.4-fold lower than in a normal heart (4289 ± 6595 vs. 23,581 ± 4074, *p* = 0.0996) (*p* < 0.001): specifically, 9-fold lower in males, *p* = 0.009, (*p* < 0.001) and 3-fold lower in females, *p* = 0.5799, (*p* < 0.001) showing, at histology, a mosaic of normal and diseased cells. M6Pr-mRNA expression was normal, while ubiquitin showed an increase of 4.6 fold vs. controls (13,284 ± 1723 vs. 2870 ± 690, *p* = 0.001) suggesting that ubiquitin-dependent post-translational degradation is likely responsible for the reduction of M6Pr in FDCM. Conclusion: M6Pr expression is remarkably reduced in FDCM as a likely result of post-translational degradation. This may explain the reduced efficacy of ERT and be a therapeutic target for the enhancement of ERT activity.

## 1. Introduction

Fabry disease (FD) is an X-linked inborn error of glycosphingolipid catabolism caused by deleterious mutations in the α-galactosidase A gene (*GLA*) encoding the lysosomal hydrolase, alpha-galactosidase A (α-Gal A) [1]. The marked deficiency or absence of α-Gal A activity results in the systemic accumulation of globotriaosylceramide (Gb-3) and related glycosphingolipids within the lysosomes, particularly in microvascular endothelial cells (ECs), vascular smooth muscle cells (VSMCs), renal tubular cells (RTCs), podocytes, and cardiomyocytes [2,3,4,5,6,7]. Their progressive accumulation, especially in ECs, podocytes, and cardiomyocytes, leads to renal failure, cardiac and cerebrovascular disease, and premature demise [1,2,3].

Currently available treatments for FD are limited and include enzyme replacement therapy (ERT, agalsidase-α and agalsidase-β since 2001 [8,9], chaperone therapy (Galactose since 2001 [10] and migalastat [11] since 2016), and a few novel approaches which are under investigation.

The efficacy of ERT on Gb-3 clearance is cell-type-specific and dose-dependent [12,13,14,15,16]. While 1 mg/kg every other week (EOW) of agalsidase-β clears endothelial and mesangial cells and fibroblasts from Gb-3 inclusions in five months, podocytes show partial clearance after one year of treatment in males with classic FD. This discrepancy may be linked to different cellular life spans, since podocytes are post-mitotic as opposed to other kidney cell types. Cardiomyocytes, similar to podocytes are post-mitotic and have long lifespans. Our knowledge on the tissue effects of ERT on cardiomyocytes is rather scarce. Using a semi-quantitative approach, Thurberg et al. [4] reported Gb-3 clearance in cardiac endothelial cells after 5 months of agalsidase-β (1 mg/kg EOW), but no clearance in cardiomyocytes. 

The reasons for the limited therapeutic impact of ERT on cardiomyocytes is still unclear but reflects negatively on patients’ outcomes.

Mannose-6-phosphate receptor (M6PR) is the major intracellular carrier of α-Gal A, and thereby, likely to affect responses to ERT and the removal of Gb-3 in Fabry cells.

Indeed, in a recent report [17], we found out that Fabry cardiomyocytes, compared with enterocytes, have remarkably reduced M6PR concentrations and fail to respond to ERT while patients’ intestinal Gb-3 is removed with symptoms resolution. This observation brought us to a wider investigation of the potential role of reduced M6PR on the resistance of FDCM to ERT administration.

Herein, M6PR was determined in endomyocardial biopsy samples from a larger cohort of patients with various GLA mutations and different severities of FDCM. Molecular processes, potentially involved with M6PR’s decline, have been addressed through the assessment of M6PR-mRNA to determine the receptor’s gene expression and quantification of Ubiquitin to test a possible involvement of post-translational M6PR protein degradation.

### Patient Population

Seventeen patients (10M, 7F, 45 ± 16) with a various severity of Fabry cardiomyopathy (Left ventricular maximal wall thickness—MWT—ranging from 11.5 to 20 mm) undergoing LV endomyocardial biopsy for diagnosis confirmation were enrolled in the study. Their clinical enzymatic, genetic, and histologic characteristics are summarized in Table 1. All patients had alpha or beta agalsidase for ≥3 years without a reduction in LV mass nor an increase in T1 mapping at CMR.

Controls were surgical biopsies from 15 patients affected by mitral stenosis with normal MWT and cardiac functions who were undergoing mitral valve replacement.

## 2. Materials and Methods

In all patients, cardiac investigations included non-invasive (ECG, Holter, 2D-echo, CMR) and invasive (coronary, left ventricular angiography and endomyocardial biopsy) studies after receiving written informed consent. Biopsy samples were processed for histology and electron microscopy. The assessment of Mannose-6-phosphate receptor (M6PR) was obtained through protein isolation, Western Blot analysis, and relative mRNA quantification.

The study complies with the Declaration of Helsinki, the locally appointed ethics committee (opinion number 6/2019) approved the research protocol, and informed consent was obtained from all subjects. 

### 2.1. Cardiac Magnetic Resonance (CMR)

CMR exams were performed on a 1.5 Tesla scanner. Standard cardiac magnetic resonance protocol included: (i) Cine magnetic resonance images acquired during breath-holds in the short-axis, 2-chamber, and 4-chamber; (ii) black blood T2-weighted short tau inversion recovery images on short-axis planes covering the entire left ventricle during 6 to 8 consecutive breath-holds for myocardial edema detection; (iii) late gadolinium-enhanced imaging performed 15 min after injection of 0.2 nmol/kg of gadoteratemeglumine and signal intensity value 2 SDs above the mean signal intensity of the remote normal myocardium were considered suggestive for myocardial fibrosis; (iv) T1 mapping imaging was also performed in all cases before and after contrast using a modified look-locker (MOLLI) sequence acquired in the short axis at basal, mid, and apical segments, before and after contrast. Global T1 value was measured as the mean of T1 values calculated in all 3 planes, avoiding LGE areas and leaving adequate margins of separation from tissue interfaces with adipose epicardial tissue and ventricular cavities. 

### 2.2. Invasive and Endomyocardial Biopsy Studies

Cardiac catheterization with left ventricular and coronary angiography was obtained in all patients. Endomyocardial biopsy (four to eight samples per patient) was performed in the septal-apical region of the left ventricle. 

### 2.3. Histology and Electron Microscopy

For histological analysis, the endomyocardial samples were fixed in 10% buffered formalin and paraffin embedded. Five-micron-thick sections were stained with hematoxylin and eosin and Masson trichrome. For electron microscopy studies, additional samples were fixed in 2% glutaraldehyde in a 0.1 M phosphate buffer at pH 7.3, post-fixed in osmium tetroxide, and processed following a standard protocol for embedding in Epon resin. Ultrathin sections were stained with uranyl acetate substitute and lead hydroxyde. 

### 2.4. Morphometric Studies 

The cardiomyocyte cross-sectional area was computed measuring the cardiomyocyte diameter across the nucleus in 50 to 100 cells in transverse sections. At that level, the diameter of the perinuclear vacuoles was also measured, and the percent cardiomyocyte area occupied by vacuoles was calculated. These measurements were analyzed using Nis-Elements BR software.

### 2.5. Assessment of Mannose-6-Phosphate Receptors and Ubiquitin in the Myocardial Tissue of Fabry Disease Patients 

We determined the expression of Mannose-6-phosphate receptors and ubiquitin in frozen myocardial tissue. Results were compared with values from surgical control unloaded myocardium (papillary muscle of patients with mitral stenosis undergoing valve replacement).

### 2.6. Protein Isolation and Western Blot

Heart tissue samples were treated as previously described [18]. The expression of Mannose-6-phosphate receptors, molecular weight 275 kDa, was visualized by using IGF-II Receptor/CI-M6PR, (D8Z3J) polyclonal antibody (1:1000, Cell Signaling Technology). The expression of ubiquitin, molecular weight 75 kDa, was evaluated using Ubiquitin Monoclonal Antibody (Ubi-1) (1:1000, Invitrogen ThermoFisher, Waltham, MA, USA 02451). Anti-α-sarcomeric actin antibody (1:500, Sigma-Aldrich, Burlington, MA, USA), molecular weight 43 kDa, was used for normalization. Signal was visualized using a secondary horseradish-peroxidase-labeled goat anti-mouse antibody (goat anti-mouse IgG-HRP 1:5000, SantaCruzBiotechnology, Dallas, TX, USA) and enhanced chemiluminescence (ECL Clarity Bio-Rad). The purity as well as equal loading (40γ) of the protein was determined by Nanodrop One (Thermofisher). To normalize target protein expression, the band intensity of each sample is determined by densitometry with Image J software. Next, the intensity of the target protein is divided by the intensity of the loading control protein. This calculation adjusts the expression of the protein of interest to a common scale and reduces the impact of sample-to-sample variation. Relative target protein expression can then be compared across all lanes to assess changes in target protein expression across samples. Digital images of the resulting bands were quantified by the Image Lab software package (Bio-Rad Laboratories, Munchen, Germany) and expressed as arbitrary densitometric units.

### 2.7. Gene Expression Study 

Patient tissues were collected from the left ventricular (LV), preserved in liquid nitrogen, and treated with Animal Tissue RNA Purification Kit (Cat. 25700) Norgen Biotek in order to extract RNA according to the manufacturer’s instructions. *β-actin* was used as reference gene. Sensi FAST Probe No-ROX One-Step Kit was used to perform the reaction. Real-time PCR was conducted with LineGene K Plus Real-Time PCR Detection System (Hangzhou Bioer Technology Co. Ltd. (BIOER).

Primer and probe for M6Pr and β-actin were designed using the RealTimeDesign™ qPCR Assay Design Software (LGC Biosearch Technologies):

*M6Pr**forward*: 5′ d TGTGGCGAGGTTCTGTTGT 3′

*M6Pr**reverse*: 5′ d TTGGGCCTCACTGCTCATC 3′

*Forward β-ACTIN*: 5′ d GGCACCCAGCACAATGAAG 3′

*Reverseβ-ACTIN*: 5′ d GCCGATCCACACGGAGTA 3′

In accordance to the Minimum Information for Publication of Quantitative Real-Time PCR Experiments, we evaluated (M6Pr and β-actin) primer efficiency [19]. The relative quantification of the M6Pr transcript was evaluated with the comparative threshold method (2^−ΔΔCt^) and Pfaffl model [20] with Excel (Office 365, Microsoft). All samples were loaded in triplicate, and an RNA pool was used as a healthy control.

### 2.8. Statistical Analysis

Statistical analysis was performed by the GraphPad Prism package, version 5.02 (GraphPad Software Inc., San Diego, CA, USA). Comparison between groups was performed with a Mann–Whitney non-parametric test. A *p* value less than 0.05 was considered statistically significant.

## 3. Results

Comparison of clinical and endomyocardial biopsy data among patient population Group A (*n* = 17) and surgical controls Group B (*n* = 15) are shown in Table 1 and Table 2.

### 3.1. Cardiac Studies

All patient populations manifested normal coronary angiography and left ventricular hypertrophy at a CMR ranging from 11.5 up to 22 mm of maximal wall thickening. Alteration of T1 mapping and LGE were commonly appreciated and are reported in Table 2.

### 3.2. Immuno-Histochemistry Study

In histology, all FDCM patients presented hypertrophied cardiomyocytes 28.3 ± 1.19 vs. controls 10.6 ± 1.8, with cytoplasmic vacuoles (mean value 38 ± 0.13%) (Figure 1A) that in electron microscopy (Figure 1B) corresponded to the accumulation of myelin bodies. Myocardial fibrosis quantified morphometrically after Masson staining was focal and mild, mean value 4.5 ± 1(%) vs. controls 2.2 ± 0.4 (%).

After immunohistochemistry characterization of inflammatory cells with monoclonal antibody anti-CD3, CD 45Ro, and CD68, no evidence of myocarditis was recorded in our patient population.

Semiquantitative evaluation of immunostaining (grades from 0 to 4) for Mannose 6 phosphate receptors showed a reduced staining (1.1 ± 1.27 vs. controls 3.4 ± 0.38) in all cardiomyocytes (Figure 1C) of male and affected myocytes of female vs. controls (Figure 1D). In contrast, endothelial cells of intramural vessels were free of Gb-3 inclusions and showed normal immunostaining (Figure 1C).

Severe reduction of MRP6 was even documented in affected cardiomyocytes of women with Fabry disease cardiomyopathy (Figure 2A–D).

### 3.3. Western Blot Analysis

Protein analysis showed M6Pr in FDCM to be 5.4-fold lower than in a normal heart (respectively 23,581 ± 4074 vs. 4289 ± 6595, *p* = 0.0996) (*p* < 0.001): specifically, 9-fold lower in males, *p* = 0.009, (*p* < 0.001) and 3-fold lower in females, *p* = 0.5799, (*p* < 0.001) showing, in histology, a mosaic of normal and diseased cells. Expression protein degradation pathway (Ubiquitin) evaluated in all patients showed an increase of 4.6 fold vs. controls (13,284 ± 1723 vs. 2870 ± 690, *p* = 0.001).

Correlation between M6Pr expression and MWT was not statistically significant (Spearman Rho = 0.423, ns)

### 3.4. M6PR m-RNA Expression

The relative quantification of M6PR patient’s mRNA highlighted no statistically significant variation (*p* = 0.4752), as resumed in Table 2. Negative control (lacking template) yielded no detectable fluorescence. The reaction efficiency evaluated is 108% and 107%, with R^2^ = 0.98 and R^2^ = 0.99 for M6Pr and β-actin, respectively. 

All cardiac histological and morpho-molecular data are shown in Table 3.

## 4. Discussion

There is general agreement on the limited efficacy of ERT in the removal of GB-3 from Fabry cardiomyocytes [4,21,22]. In this regard, various hypotheses have been advanced, including interstitial expansion by myocardial fibrosis [23,24,25,26], auto-reactive myocardial inflammation in response to GB-3 release [27], and an insufficient dose of α/β-agalsidase administration [28]. Up to now, no definitive explanation has been provided, and the progression of FDCM still remains unaffected, particularly for the advanced forms. The present study explored the potential pathogenetic role of M6Pr being the major cellular mechanism of agalsidase uptake and targeting to lysosomes. Results of the present study suggest a consistent downregulation of myocardial M6Pr, with a mean value reduction of 4.6 fold vs. controls (Figure 3C) and a variance between the male (reduced by 9 fold, Figure 3D) and female (diminished by 3 fold, Figure 3E) FDCM population. This sex discrepancy is explained by the X-linkage of the GLA gene and the presence in the female myocardium of normal and diseased myocytes (cell mosaic) (Figure 2A) as an expression of skewed X-chromosome inactivation. Normal immunostaining for M6Pr of unaffected myocytes (Figure 2C) in FDCM females, as well as of GB-3 free microvascular endothelial cells in all FDCM populations, confirm the importance of M6Pr in promoting cell entrance and the targeting of ERT.

On the other hand, among the other factors potentially interfering with ERT delivery, myocardial fibrosis having a focal and mild resistance to therapy was unjustified, while myocardial inflammation creating a barrier of edema and autoreactive immunocytes was not a finding.

These observations make unlikely the solution of ERT resistance of FDCM only with an increase in enzyme administration while points to the pharmacological possibilities to enhance M6Pr expression in cardiomyocytes [29]. In this regard, hormonal agents such as growth hormone and estradiol [30] have been recognized as being able to increase cellular M6Pr expression. Combining ERT with M6Pr-enhancing agents, this may potentially improve the mobilization of GB-3 and FDCM outcomes.

As far as the mechanism of M6Pr’s decline in myocardiocytes is concerned, this is still unclear.

The normal level of M6Pr-mRNA obtained in our patients population suggests GLA gene expression as not being involved.

In the present study, we tested the hypothesis that normally synthetized M6Pr might be polyubiquinated in the cytoplasm of cardiomyocytes and then degraded. Indeed, it is known that polyubiquinated substrates are transported to proteasomes, recognized by a receptoral component of the proteasome, deubiquinated, pass into the 20S proteasomal unit, and are unfolded and cleaved, generating small peptides which are released into the cytosol for recycling and further disposal [31]. In our cohort of patients with FDCM not responding to ERT administration, we found out a marked elevation of myocardial ubiquitin vs. controls, indicating a possible post-translational degradation of the receptor. (Figure 4). Specifically, it may be the case that cytosolic GB-3 might link M6Pr after its synthesis, promoting proteasome formation and protein degradation. If this pathway will be confirmed, it would disclose another potential source of enhancement for ERT’s impact in FDCM. Indeed, recent advances have shown that proteasomal functions may be inhibited/regulated by drugs including monoclonal antibodies and small molecules. Bortezomib was the first monoclonal antibody approved by the US FDA as a proteasome inhibitor for therapy of multiple myeloma, solid tumors, and some inflammatory diseases [31]. Other similar inhibitors include Carfilzomib, Ixazomib, and small molecule drugs actually under development.

Therefore, in patients showing polyubiquinated M6Pr, a treatment with proteasomal inhibitors could at least partially restore their function, enhancing ERT efficacy.

The caveat to the administration of proteasomal inhibitors is the presence or the development of cardiovascular events including arrhythmias, ischemic heart disease, hypertension, or heart failure [32]. Close patient monitoring during their use is highly recommended.

### 4.1. Clinical Implications

FD cells with low turnover as cardiomyocytes and podocytes appear to be more resistant to ERT impacts compared with high-turnover cells such as endothelial [4] and intestinal cells [17] where a clinical improvement and a pathological clearance of GB-3 has been demonstrated after ERT administration. It is possible that even podocytes, similarly to cardiomyocytes, have reduced M6Pr expression, and that this might be the biological basis of renal resistance to ERT. If combination therapy with ERT and enhancing M6Pr-drugs or proteasomal inhibitors will bring FDCM improvement, even Fabry renal disease has chances to be ameliorated.

### 4.2. Limitations

Unclarified the translational degradation of M6Pr.

### 4.3. Translation Perspective

In addition to ERT, inductors of M6Pr such as GH or estradiol may enhance the ability of intracellular transportation of infused enzymes, obtaining an increased mobilization of GL-3 from cardiomyocytes and the improvement of patients’ treatment and outcomes.

## 5. Conclusions

M6Pr expression is remarkably reduced in the myocardium of patients with FDCM as a likely result of post-translational degradation. This may explain the reduced efficacy of ERT and be a potential therapeutic target for the enhancement of ERT activity.

## Abbreviation

ERTEnzyme replacement therapyFDFabry diseaseCMRCardiac magnetic resonanceLVLeft ventricularFDCMFabry Disease CardiomyopathyGB-3GlobotriaosylceramideMWTMaximal wall thicknessM6PrMannose-6-phosphate receptors

## Figures and Tables

**Figure 1 jcm-11-05440-f001:**
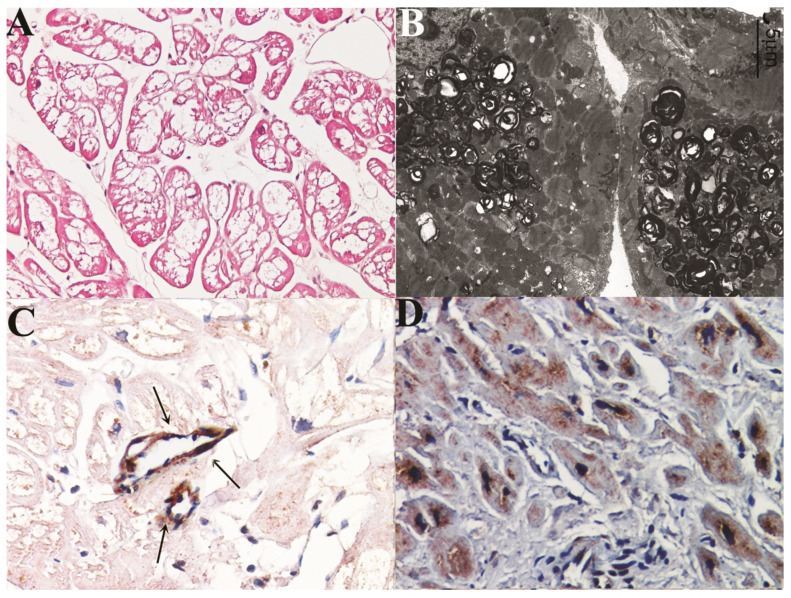
Histology (**A**), electronmicroscopy (**B**) and immunohistochemistry (**C**) for M6Pr in a 43-year-old man (pt 1) affected by Fabry disease cardiomyopathy (MWT 14 mm). In panel (**C**), normal immunostaining of endothelial cells, (black arrows) in contrast with reduced staining of cardiomyocytes, is observed. Panel (**D**) represents immunohistochemistry for M6Pr in normal myocardium. The pictures suggest a severe reduced expression of M6Pr in Fabry cardiomyocytes vs. controls.

**Figure 2 jcm-11-05440-f002:**
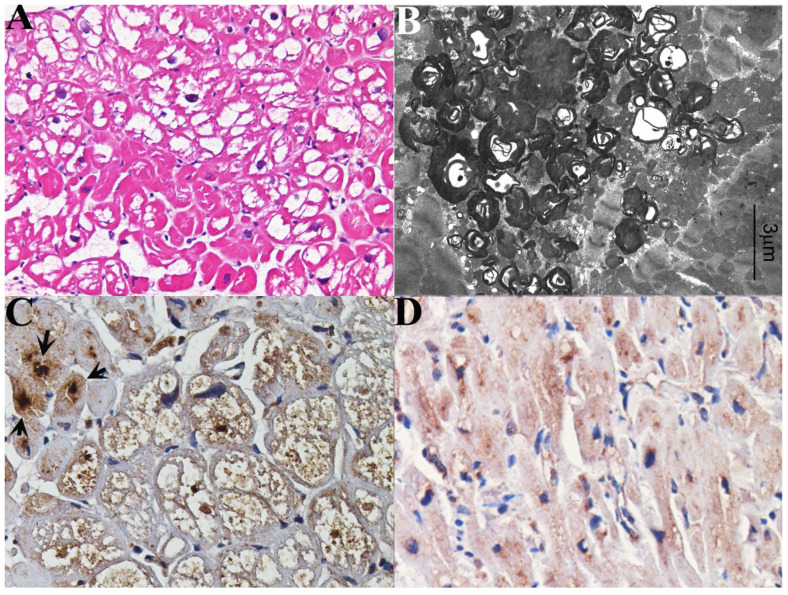
Histology (**A**), electronmicroscopy (**B**) and immunohistochemistry (**C**) for M6Pr in a 54-year-old woman (pt 4) and immunohistochemistry for M6Pr in normal myocardium (**D**). Comparison of the two pictures (**C**) vs. (**D**) suggest a severe reduction in M6Pr expression in Fabry cardiomyocytes. Of note in (**C**), due to patchy cardiomyocyte involvement, unaffected cells (see arrows) are strongly positive as compared to the affected cells.

**Figure 3 jcm-11-05440-f003:**
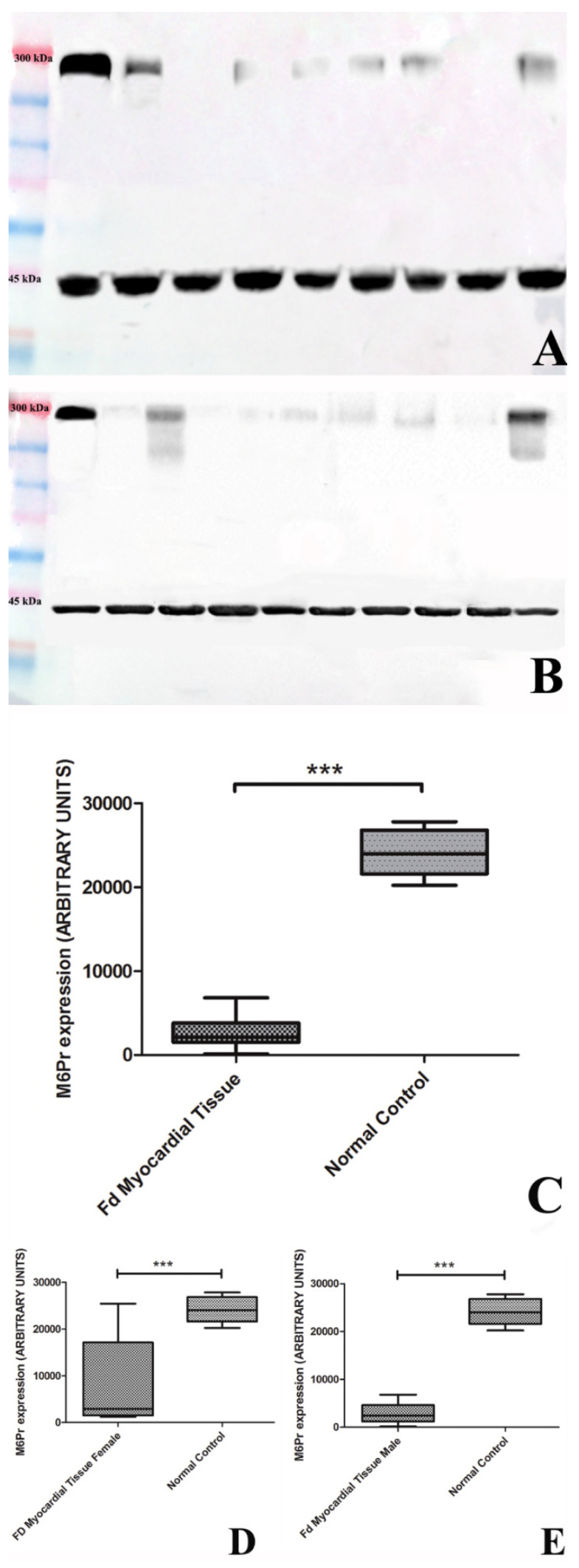
Panel (**A**,**B**): Western blot analysis of all 17 patients of M6Pr, molecular weight 275 Kd. Panel (**A**): Lane1 = Markers; Lane 2 = Control; Lane 3 = patient 1; Lane 4 = patient 2; Lane 5 = patient 3; Lane 6 = patient 4; Lane 7 = patient 5; Lane 8 = patient 6; Lane 9 = patient 7; Lane 10 = patient 8. Panel (**B**): Lane 1 = Markers; Lane 2 = Control; Lane 3 = patient 9; Lane 4 = patient 10; Lane 5 = patient 11; Lane 6 = patient 12; Lane 7 = patient 13; Lane 8 = patient 14; Lane 9 = patient 15; Lane 10 = patient 16; Lane 11 = patient 17. Panel (**C**–**E**): Graphs document western blot of M6Pr, in all 17 patients with FDCM showing 5.4-fold lower of M6Pr values in patients versus normal heart (4289 ± 6595 vs. 23,581 ± 4074, *p* = 0.0996) (*p* < 0.001), (**C**), 9-fold lower in male, *p* = 0.009, (*p* < 0.001), (**D**) and 3-fold lower in female *p* = 0.5799, (*p* < 0.001), (**E**). Alpha sarcomeric actin (43 kDa) was used as a loading control. *** = *p* < 0.001.

**Figure 4 jcm-11-05440-f004:**
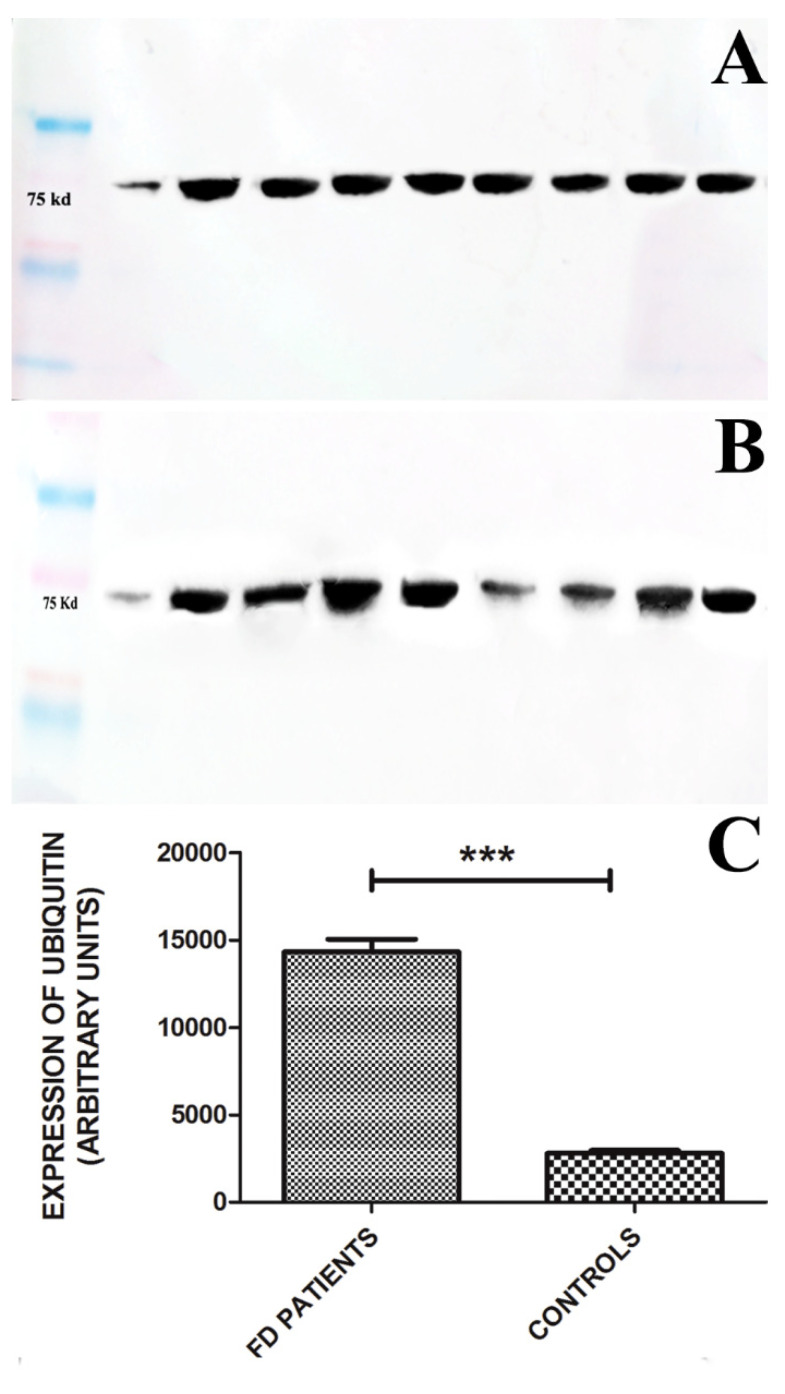
Panel (**A**,**B**): Western blot analysis of Ubiquitin in 16 out of 17 patients with FDCM, molecular weight 75 Kd. Panel (**A**): Lane1 = Markers; Lane 2 = Control; Lane 3 = patient 1; Lane 4 = patient 2; Lane 5 = patient 3; Lane 6 = patient 4; Lane 7 = patient 5; Lane 8 = patient 6; Lane 9 = patient 7; Lane 10 = patient 8. Panel (**B**): Lane 1 = Markers; Lane 2= Control; Lane 3 = patient 9; Lane 4 = patient 10; Lane 5 = patient 11; Lane 6 = patient 12; Lane 7 = patient 13; Lane 8 = patient 14; Lane 9 = patient 15; Lane 10 = patient 16; Panel (**C**): Graphs report western blot of Ubiquitin in 16 out of 17 patients and show 4.6-fold increase in Ubiquitin values vs. controls (13,284 ± 1723 vs. 2870 ± 690, *p* < 0.001). *** = *p* < 0.001.

**Table 1 jcm-11-05440-t001:** Baseline clinical and genetic features of patient population.

	Age/ Sex	GLA Mutation	αGAL- A Enzymatic Activity nmol/h/mL	Extra-Cardiac Features	Cardiac Symptoms	Enzymatic Replacement Therapy	ERT Duration (Months) at Enrollment
Pt 1	42/M	c.666delC	0	cornea verticillata, gastrointestinal disorders, weight loss, proteinuria, hypo-anhidrosis	dyspnea, palpitations	agalsidase alpha	36
Pt 2	27/M	c.666delC	0	gastrointestinal disorderders, weight loss	palpitations	agalsidase alpha	36
Pt 3	22/F	c.666delC	9.2	gastrointestinal disorderders, weight loss, cornea verticillata	palpitations	agalsidase alpha	36
Pt 4	58/F	c.668G > A p.C223Y	3.3	cornea verticillata, angiokeratomas, hypo-anhidrosis, acroparesthesias	dyspnea, palpitations	agalsidase alpha	40
Pt 5	52/F	c.547+ 1G > A	3.4	proteinuria, cornea verticillata	dyspnea, palpitations	agalsidase alpha	38
Pt 6	27/M	deletion exons 3 and 4	0	cornea verticillata, angiokeratomas, hypo-anhidrosis, acroparesthesias	palpitations	agalsidase alpha	42
Pt 7	52/F	deletion exons 3 and 4	3.5	acroparesthesias, sicca syndrome, arthritis, proteinuria	dyspnea, palpitations	agalsidase alpha	39
Pt 8	22/M	c.644A > G	0.3	none	palpitations	agalsidase alpha	36
Pt 9	62/M	c.644A > G	0	cornea verticillata, microalbuminuria	dyspnea	agalsidase alpha	36
Pt 10	66/M	c.644A > G	0	cornea verticillata, proteinuria	dyspnea, palpitations	agalsidase alpha	40
Pt 11	36/F	C.548 > G	3.8	cornea verticillata, proteinuria	dyspnea, palpitations	agalsidase alpha	39
Pt 12	54/M	c.644A > G	0	cornea verticillata	dyspnea, palpitations	agalsidase alpha	36
Pt 13	30/M	deletion exons 3 and 4	0	cornea verticillata, angiokeratomas, hypo-anhidrosis, acroparesthesias, proteinuria	dyspnea, palpitations	agalsidase beta	36
Pt 14	55/F	deletion exons 3 and 4	3.1	acroparesthesias, cornea verticillata, proteinuria	dyspnea	agalsidase beta	38
Pt 15	37/F	C.548 > G	4.0	cornea verticillata, proteinuria	dyspnea	agalsidase alpha	36
Pt 16	59/M	c.644A > G	0	cornea verticillata	dyspnea, angor	agalsidase alpha	40
Pt 17	66/M	c.644A > G	0.4	acroparesthesias, cornea verticillata	dyspnea, palpitations	agalsidase alpha	39

GLA, α-galactosidase A gene; αGAL- A, alpha galactosidase A; ERT, Enzymatic replacement therapy.

**Table 2 jcm-11-05440-t002:** Baseline instrumental features of patient population.

	Age/ Sex	ECG	Diastolic Function 2d-Echo	LV Mass/BSA g/m^2^ (CMR)	MWT mm (CMR)	LV EDV/BSA mL/m^2^ (CMR)	LV ESV/BSA (mL/m^2^) (CMR)	LVEF (%) (CMR)	T1 Mapping ms (CMR)
Pt 1	42/M	short PR, LVH	grade I	112.0	14	106.3	33.9	68.1	874
Pt 2	27/M	short PR	grade I	28.1	10	82.7	31.8	61.4	860
Pt 3	22/F	PSVC	normal	26.7	9	50.1	20.3	59.4	980
Pt 4	58/F	PSVC, PVC, PAF	grade I	63.6	16	66.5	27.7	59.7	Not detected
Pt 5	52/F	short PR, LVH	grade II	99.1	15	86.9	23.8	68.0	Not detected
Pt 6	27/M	short PR	grade I	58.4	11.5	72.1	26.1	63.9	860
Pt 7	52/F	1st degree AV block	grade I	53.1	10	78.1	21.1	72.9	940
Pt 8	22/M	short PR	grade I	60.7	10	100	43.2	57	930
Pt 9	62/M	bifascicular block	grade II	146.9	25	108	54.4	49.7	860
Pt 10	66/M	LBBB	grade II	190.6	20	142.8	79.8	44.1	870
Pt 11	36/F	short PR	grade I	61.1	11	85.4	27.8	67.0	990
Pt 12	54/M	LVH	grade I	142.6	26	79.7	31.7	60.1	912
Pt 13	30/M	short PR, PSVC	grade I	62.3	12	58.3	22.3	61.7	910
Pt 14	53/F	1st degree AV block	grade I	58.0	11	74.0	23.0	69.0	940
Pt 15	37/F	short PR, PSVC	grade I	50.5	11	97.1	35.1	63.0	990
Pt 16	59/M	LBBB	grade II	165.5	28	81.6	31.3	61.5	930
Pt 17	66/M	LBBB	grade III	168.9	28	102.7	54.7	47	850

LVH, left ventricular hypertrophy; PSVC, premature supraventricular contractions; PVC, premature ventricular contractions; PAF, paroxysmal atrial fibrillation; AV, atrioventricular; LBBB, left bundle branch block; CMR, cardiac magnetic resonance; LV, left ventricular; BSA, body surface area; MWT, maximal wall thickness; LVEF, left ventricular ejection fraction; Alpha-galactosidase A enzymatic activity normal if >3 nmol/h/mL. T1 mapping normal if >980 ms. ECG = Electrocardiogram; EDV = End diastolic volume. Short PR = ECG parameter_PR = Depolarization of Atrial Myocardium.

**Table 3 jcm-11-05440-t003:** Cardiac histologic and morpho-molecular findings in Fabry patients (Group A) vs. normal controls (Group B).

	Group A	Group B	*p* Value
Cardiomyocytes diameter (µm)	28.3 ± 1.19	10.6 ± 1.8	*p* < 0.001
Extent of myocardial fibrosis (%)	4.5 ± 1	2.2± 0.4	*p* < 0.001
Mannose expression (IHC grading)	1.1 ± 1.27	3.4 ± 0.38	*p* < 0.001
Mannose 6-phosphate Receptor Western Blot (Arbitrary Units)	4289 ± 6595	23,581 ± 4074	*p* < 0.001
Protein degradation pathway_ Ubiquitin (Arbitrary Units)	13,284 ± 1723	2870 ± 690	*p* < 0.001
Mannose 6-phosphate Receptor m-RNA expression	1.13 ± 0.43	1.01 ± 0.12	*p* = 0.4752 (ns)

*p* values referred to comparison between two groups. *p* value < 0.05 was considered statistically significant. Quantitative measurements are expressed as mean ± SD. ns = not significant. IHC = Immunohistochemical grading.

## Data Availability

The datasets used and analyzed during the current study are avail-able from the corresponding author upon reasonable request.

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
