# Peer review of "Downregulation of Mannose-6-Phosphate Receptors in Fabry Disease Cardiomyopathy: A Potential Target for Enzyme Therapy Enhancement"

_jcm, 2022, doi:10.3390/jcm11185440_

Round 1

Reviewer 1 Report

The manuscript titled “Down Regulation of Mannose-6-Phosphate Receptors in Fabry Disease Cardiomyopathy. Potential Target for Enzyme Therapy Enhancement” is extremely interesting. However, I would put forward a few suggestions towards further improving the conclusions drawn by the manuscript

.

Major points:

1.     The different degree of cardiomyopathy as evident from echocardiographic data and ECG data might be correlated with expression of Mannose-6-Phosphate Receptors,

2.     Comparison of fibrosis of the three myocardium layers of the anterior wall, lateral wall, posterior wall of the left ventricle, interventricle septum and the right ventricle.

3.     The involvement of Mannose 6 phosphate receptors with renin pathway in cardiac muscle has previously been documented. (https://doi.org/10.1161/01.HYP.30.6.1389)

The study should attempt to provide a positive correlation with the role of M6PR and cardiomyopathy.

4.     Lastly, the status of inflammatory molecules expression from patient tissue and blood concentration of inflammatory proteins like TGF beta, TNF alpha etc.

Author Response

  1. The different degree of cardiomyopathy as evident from echocardiographic data and ECG data might be correlated with expression of Mannose-6-Phosphate Receptors,

Reply: We tried to correlate maximal Left ventricular wall thickness (MWT) with M6Pr expression. In the text (Page 7, line 213-214) we reported that Correlation between M6Pr expression and MWT was not statistically significant (Spearman Rho=0.423,ns)

  1. Comparison of fibrosis of the three myocardium layers of the anterior wall, lateral wall, posterior wall of the left ventricle, interventricle septum and the right ventricle.

Reply: Myocardial fibrosis expressed at cardiac MRI as LGE was tipically more prominent in the infero-lateral wall of the left ventricle.

  1. The involvement of Mannose 6 phosphate receptors with renin pathway in cardiac muscle has previously been documented. (https://doi.org/10.1161/01.HYP.30.6.1389)

The study should attempt to provide a positive correlation with the role of M6PR and cardiomyopathy.

Reply: Correlation between severity of Fabry cardiomyopathy and reduced expression of M6PR is very difficult to be obtained because of extreme sampling variability : number of normal vs affected cardiac cells in woman, extent of myocardial fibrosis may remarkably vary among biopsy samples. This point could be clarified further selecting through Laser capture microdissection cardiomyocytes with different degree of hypertrophy and GB3 accumulation.

  1. Lastly, the status of inflammatory molecules expression from patient tissue and blood concentration of inflammatory proteins like TGF beta, TNF alpha etc.

Reply: We failed to assess inflammatory proteins following the histologic observation of absence of myocardial inflammation.

Reviewer 2 Report

Frustaci et al., show down-regulation of mannose-6-phosphate receptors (M6Pr) in Fabry 2 Disease Cardiomyopathy (FDCM). They also suggest that ubiquitin-dependent post-translational degradation is likely responsible for the reduction of M6Pr in FDCM. The current study is interesting and relevant to the field of cardiovascular diseases. However, this reviewer has a few concerns that need to be addressed before publication.

1. Author may need to state the hypothesis in the abstract section.

2. Results section needs to be properly modified based on the figures present in the manuscript. It's really confusing for the reader to understand the study. Each of the panels in the figures needs to be stated. For example, what was the purpose of panels A, B, C, and D in figure 1? 

3. Where is the description of figure 2 in the result section?

4. Figure 3A and 3B: The author may need to write the protein name for the western blot bands at 300kDa and 45kDa.

5. Where are the loading control for the western blots in Figure 3?

6. M6Pr protein expression quantitation in Figure 3 does not correlate with the result in Table 3. 

7. It would be great if the author can include the ubiquitin western blot image in the main figure along with its quantitation. 

Author Response

Frustaci et al., show down-regulation of mannose-6-phosphate receptors (M6Pr) in Fabry 2 Disease Cardiomyopathy (FDCM). They also suggest that ubiquitin-dependent post-translational degradation is likely responsible for the reduction of M6Pr in FDCM. The current study is interesting and relevant to the field of cardiovascular diseases. However, this reviewer has a few concerns that need to be addressed before publication.

  1. Author may need to state the hypothesis in the abstract section.

Reply: As suggested the working hypothesis on the mechanism of MRP6 is included now in the abstract.

  1. Results section needs to be properly modified based on the figures present in the manuscript. It's really confusing for the reader to understand the study. Each of the panels in the figures needs to be stated. For example, what was the purpose of panels A, B, C, and D in figure 1?

Reply: The meaning of Figure 1 and 2 have been reported and clarified in the results section (page 6, line 198-206)

  1. Where is the description of figure 2 in the result section?

Reply: Description of Fig 2 has been inserted in the results section (page 6, line 207-208)

  1. Figure 3A and 3B: The author may need to write the protein name for the western blot bands at 300kDa and 45kDa.

Reply:

As suggested the protein name and molecular weight has been added in line 248 and line 255

  1. Where are the loading control for the western blots in Figure 3?

Reply:

Alpha sarcomeric actin (43 kDa) was used as loading control (pag. 10,line 257)

  1. M6Pr protein expression quantitation in Figure 3 does not correlate with the result in Table 3.

Reply:

I apologize for the mistake, the appropriate number has been placed in Table 3

  1. It would be great if the author can include the ubiquitin western blot image in the main figure along with its quantitation.

Reply: Thank you for your suggestion: western blot of myocardial samples from 16 of our patient population are now provided in an additional Figure (Fig 4).

Round 2

Reviewer 1 Report

The authors failed to satisfy the queries raised.

Reviewer 2 Report

The authors have been very responsive to the queries. This manuscript now can be accepted.